# Numerical Analysis and Recursive Compensation of Position Deviation for a Sub-Millimeter Resolution OFDR

**DOI:** 10.3390/s20195540

**Published:** 2020-09-27

**Authors:** Yueying Cheng, Mingming Luo, Jianfei Liu, Nannan Luan

**Affiliations:** 1School of Electronic and Information Engineering, Hebei University of Technology, Tianjin 300401, China; 201821902035@stu.hebut.edu.cn (Y.C.); 2019013@hebut.edu.cn (M.L.); luan@hebut.edu.cn (N.L.); 2Tianjin Key Laboratory of Electronic Materials and Devices, Tianjin 300401, China; 3Hebei Key Laboratory of Advanced Laser Technology and Equipment, Tianjin 300401, China

**Keywords:** optical frequency domain reflectometry, position deviation compensation, sub-millimeter spatial resolution

## Abstract

We analyze the source of the position deviation and propose a demodulation recursive compensation algorithm to ensure a sub-millimeter resolution in improved optical frequency domain reflectometry. The position deviation between the geometric path and optical path changes with the temperature or strain, due to the elastic-optic and thermal-optic effects. It accumulates along the fiber and becomes large enough to affect the spectral correlation between the measured and reference spectra at the fiber end. The proposed algorithm compensates for the position deviation of each measuring point and aligns the measured spectra with its reference. The numerical and experimental results both reveal that the signal-to-noise ratio of the correlation is improved doubly and a sub-millimeter spatial resolution becomes available at a 30 m fiber end. The recursive compensation algorithm helps to restrain the correlation degeneration at the fiber end and promises an effective approach to a sub-millimeter resolution in optical frequency domain reflectometry.

## 1. Introduction

Optical frequency domain reflectometry (OFDR), as a promising technique based on intrinsic Rayleigh scattering (RS), was firstly introduced by W. Eickhoff in 1981 [1]. Initially, OFDR was mainly used for loss and breakpoint diagnosis in optical fiber devices and networks [2,3]. The spatial resolution can be achieved at several micrometers in the frequency domain [4,5], with an expandable measuring length to tens or hundreds of meters. With the development of the narrow linewidth tunable laser source (TLS) and nonlinear phase noise compensation in OFDR, the sensing range can be promoted to dozens of kilometers [6,7,8,9]. In addition, Froggatt et al. proposed a distributed quantitative sensing method with a high spatial resolution of 0.6 cm and sensitivity of 5 micro-strains over 30 cm, utilizing spectral correlation shifts in OFDR [10,11]. This method is suitable for practical applications, where the higher spatial resolution and longer sensing range is essential, such as in temperature sensing in nuclear reactors, deformation monitoring of wind turbines, and wing skin monitoring of aircraft [12,13,14]. However, the similarity of RS spectra degenerates at the fiber end, which limits the spatial resolution in the long detection range. Feng K. et al. demonstrated that similarity of the RS spectra degrades in the event that a large strain is loaded. They also found highly similar characteristics of local spectra and proposed a novel method to obtain wavelength offset, relying on matching measured and local reference spectra. New evaluation coefficients of spectral similarity were illustrated, based on minimum residual sum of squares and least square, respectively [15,16]. This technology significantly helps to maintain spectral signal-to-noise ratio (SNR). They obtained a 3000 micro-strain distribution curve along a 3 m sensing fiber with a 3 mm long fiber gauge. Then, Zhao S. et al. [17] proposed spectrum registration to restrain the similarity degeneration of RS spectra and ensure a high spatial resolution, using the narrow box window to extract a local spectrum in the case of a large measurable range. The combination of spatial calibration and spectrum registration can avoid the mismatch of the fiber segments. Distributed 7000 micro-strain along a 1.2 m sensing fiber is demodulated over a 5 mm spatial resolution accurately. In addition, this team also proposed a demodulation approach based on image processing to remove the noise caused by data acquired and realize accurate measurements. This technology was verified at the spatial resolution of 0.4 mm with a 350 mm sensing distance [18]. Luo M. et al. [19] realized a spatial resolution as an order of 0.5 mm over a 25 m long sensing gauge by the compensation algorithm for position deviation. They demonstrated that position deviation accumulating along the fiber degrades the similarity of the RS spectra, due to the elastic-/thermal-optic effects. However, they did not quantitatively analyze the causes, effects, and compensation mechanisms of position deviation in detail.

In this paper, aiming at the position deviation in OFDR strain sensing system, we quantitatively analyze the influence on the system. Then, the recursive compensation mechanism for position deviation is introduced in detail. This method can align the measured spectra with their reference and preserve the similarity between the measurement and reference spectra at a high spatial resolution. The experimental results indicate that the SNR of correlation is improved doubly, and the fake peaks can be efficiently eliminated. Distributed strain is demodulated with a 5 mm spatial resolution over a 30 m long detection range finally. The homogeneity between theoretical analysis and experimental verification demonstrates that this proposed technique can satisfy the requirements of high-precision distributed sensing.

## 2. Principles

Figure 1 illustrates the schematic diagram of the dual-polarization harvesting OFDR, which consists of a TLS, a reference Michelson interferometer (RMI), a heterodyne coherent detection module, and a data acquisition-processing module. The beat signal from the RMI provides an external reference clock to resample the signals and corrects the scanning nonlinearity of the TLS. Due to the optical path difference between the fiber under test (FUT) and the local reference, the reflected light interferes with the local light and generates a position dependent signal. In the event that the axial strain or temperature changes, the optical path changes along with the length and the refractive index due to the elastic-optic and thermal-optic effects, resulting in Rayleigh spectra shift. According to the correlation calculation in the spatial domain, linear temperature and strain dependent wavelength shift can be observed. Meanwhile, position deviation is introduced in the demodulation between the measured and reference spectra as well. This causes spectral dislocation and correlation degeneration, especially at the fiber end.

The position deviation is discussed in detail, as shown in Figure 2. The orange dots represent the Rayleigh scattering center of the fiber with strain or temperature applied. It includes four equivalent measuring points, with Δx deviation increment in optical path for each one. Besides, the position deviation accumulates along the fiber and increases to 4 Δx through the stretched fiber segment. The effect for position deviation is similar in strain and temperature sensing systems. In this paper, the position deviation only is analyzed in the static strain sensing system. In theory, the accumulation is expressed with the coeffects on the optical path, which is described in integral form, as shown in Equation (1). Meanwhile, in the practical applications, all signals are digitized and discretized in the sum form, as shown in Equation (2).
(1)$$X_{deviation}=\int_{0}^{x}(\kappa_{eoc}\cdot \Delta S_{dx}+\kappa_{toc}\cdot \Delta T_{dx})dx$$
(2)$$X_{deviation}=\sum_{0}^{i}(\kappa_{eoc}\cdot \Delta S_{x_{i}}+\kappa_{toc}\cdot \Delta T_{x_{i}})\Delta x_{i}$$

The $\kappa_{eoc}$ and $\kappa_{toc}$ are the elastic-optic and thermal-optic coefficient, respectively; $\Delta S_{x_{i}}$ and $\Delta T_{x_{i}}$ refer to the strain and temperature variations on a $\Delta x_{i}$ long fiber segment.

As shown in Figure 3, different axial strain stages are loaded along the 30 m long FUT. The slope of the accumulation in position deviation (brown line) strictly agrees with the strain distribution (olive line), which achieves 0.491 mm at the fiber end. The deviation seems negligible to the whole sensing fiber, but it approaches the 0.5 mm gauge length of the sensor and causes spectral dislocation, which indicates the difficulty of achieving a high spatial resolution at the fiber end.

For different spatial resolutions, the spectral dislocation caused by the 0.491 mm position deviation is approximately 25% (at 2 mm), 50% (at 1 mm), and 62.5% (at 0.8 mm). The SNR is used as an indicator to quantify the correlation between the measured and reference spectrum. As depicted in Figure 4a, with the increasing deviation between the measured and the reference spectra, the characteristic peak gradually submerges into the noise. The threshold of SNR, ensuring signal contrast, was defined as 4.59 dB (e^−1^ times of the SNR with no deviation). Figure 4b shows that the SNR of the correlation declines with the position deviation increment. In the event that the deviation reaches ~60% of the sensor gauge length, the SNR cannot satisfy the threshold, and the position deviation should be compensated to eliminate the fake peaks in correlation.

For a typical OFDR sensing system with limited data capacity, a higher spatial resolution means a weaker cross-correlation and a lower tolerance for spectral dislocation. Therefore, to achieve a higher spatial resolution at a longer fiber end, the compensation for position deviation is supposed to rematch the measured spectrum to its reference and ensure the spectral correlation with a recursive algorithm.

OFDR can be regarded as the cascaded fiber Bragg gratings (FBGs) with random period. Once the strain or temperature is applied on the so-called FBGs, the central wavelength $\lambda_{B}$ shows a strain or temperature dependent wavelength shift. By analogy with the FBG, the elastic- and thermal- coeffects on the wavelength shift are expressed by Equation (3).
(3)$$\frac{\Delta (nx_{i})}{nx_{i}}\approx \frac{\Delta n\cdot x_{i}}{nx_{i}}+\frac{n\Delta x_{i}}{nx_{i}}=\frac{\Delta n}{n}+\frac{\Delta x_{i}}{x_{i}}\approx \frac{\Delta \lambda }{\lambda_{c}}$$
where n and $x_{i}$ refer to the refractive index of fiber core and the fiber segment of $i_{th}$ sensor. The Δn and $\Delta x_{i}$ represent the refractive index and optical path changes, respectively, while $\lambda_{c}$ and $\Delta \lambda_{c}$ refer to center wavelength and wavelength shift. Therefore, the position deviation can be calculated with previous i sensors using integral Equation (4) and discrete Equation (5), respectively.
(4)$$X_{deviation}^{i+1}=\int_{0}^{l_{0}}\frac{\Delta \lambda_{i}}{\lambda_{ic}}ndx$$
(5)$$X_{deviation}^{i+1}=\sum_{0}^{i}\frac{\Delta \lambda_{i}}{\lambda_{ic}}n\Delta x_{i}$$

During the compensation process, the starting point of the measured spectrum is aligned with its reference, with the correlation well improved. According to the recursive algorithm analyzed above, the compensation for position deviation makes a high spatial resolution possible at the fiber end.

Assuming no deviation at the initial position of the stretched fiber segment. The position deviation of the second measuring point is corrected by the first point. With recursive compensation, the position deviation at the measuring point i+1 is corrected by previous i points, and the position deviation is corrected from one fiber segment to another, from the beginning to the end. The logical process based on the recursive compensation algorithm for position deviation is illustrated in Figure 5. Firstly, the spectrum at the $x_{i}$ fiber segment is selected and extracted by a sliding window, where the position deviation is corrected with the previous i−1 sensors. Then, the wavelength offset of the current sensor Δλ is obtained by correlation demodulation, which can be transferred into optical path variation by $\Delta \lambda /\lambda_{c}$ Utilizing Equation (5) to calculate the position deviation accumulated to the $x_{i}+1$ fiber segment, correlation demodulation is then performed again to get accurate strain value. Finally, the above process is repeated until the last fiber segment. Experimental verification is executed to prove the effectiveness of the above recursive compensation algorithm.

## 3. Results and Analysis

Figure 6 shows the dependence of the spectral shift on strain at a single measuring point. The blue and red lines in Figure 6a,b are the cross-correlation respectively referring to the measured spectrum without/with strain. Figure 6a is obtained by direct solution in time domain, and Figure 6b is calculated utilizing convolution of frequency domain. Whatever method is used, the blue peaks are located at the origin in the middle, while the red peaks show the offset towards the positive spectral range. By comparing Figure 6a,b, the direct solution to the cross-correlation increases the computation and complexity of the system. Besides, a large baseline, as seen in Figure 6b, also affects the signal contrast and peak-seek accuracy. Thus, we use the convolution in frequency domain to obtain wavelength shift.

The recursive compensation algorithm for position deviation is demonstrated with a static strain verification in the experiment. With massive data harvested from the experiment in frequency domain, the distributed spectra are mapped to their positions determined by the time delay. Particularly, the measured and reference spectra of the fiber segment at 21.24 m are selected as a verification for the position-deviation compensation algorithm. The SNR of the correlation peak is used as an indicator to describe the spectral correlation at different spatial resolution at 2, 1, and 0.5 mm. According to the theoretical analysis above, a higher spatial resolution means excessive data segmentation and insufficient spectral reconstruction. Thus, the same position deviation becomes more obvious and leads to a serious degeneration in correlation at a higher spatial resolution. By comparing Figure 7a,b, the position deviation causes correlation degeneration at a spatial resolution of 2 mm. Nevertheless, such position deviation is not large enough to depress the spectral contrast and affect the peak-seek accuracy. After compensation, the SNR of correlation is improved doubly. For a higher spatial resolution at 1 mm, the same deviation leads to significant correlation degeneration in Figure 7c. Even outliers are recorded as well by mistake due to low contrast during the peak-seek, unless the deviation is well corrected and the fake peaks are removed, as in Figure 7d. For extreme cases at a 0.5 mm spatial resolution, the spectral correlation is totally lost in Figure 7e, leaving the strain demodulation impossible at such a high resolution. However, with the position-deviation compensation in Figure 7f, the characteristic peak appears where it should be. Since a higher spatial resolution means a narrower sliding window, the data capacity of correlation calculation becomes smaller. In the event that the data capacity approaches its limit, improvement can hardly be achieved in SNR at high spatial resolution. Experimentally, the recursive algorithm is proven to be effective through the comparison of correlation SNR with and without the position-deviation compensation. Moreover, the alignment between the measured and reference spectra promises the possibility of a high spatial resolution at the fiber end.

A static strain is axially applied at the end of a 30 m long optical fiber in an experimental verification. Figure 8 shows the strain distribution curves along an optical fiber segment at different spatial resolutions. The correlation SNR still remains considerable at 2 mm spatial resolution, while it decreases down to the background noise at 0.5 mm spatial resolution. In this case, the fake peaks randomly appear anywhere along the correlation curve, and the fake strain is obtained with the fake peak shift. Once the deviation becomes significant at a 0.5 mm spatial resolution, the errors randomly scatter near the strain distribution curve in Figure 8b. With the recursive compensation, the random errors are effectively suppressed and the restored strain distribution curve is close to that at 2 mm spatial resolution. Thus, the recursive algorithm for position deviation compensation is proven to be effective to restrain the correlation degeneration and improve SNR of the characteristic peak. The approach to a sub-millimeter spatial resolution is theoretically analyzed and demonstrated with a verification of 0.5 mm. Besides, the reorganization of 0.5 mm fiber segment is practically achieved in the experiment, referring to the experimental results in [19].

## 4. Conclusions

We quantitatively analyze the position deviation and demonstrate that it degrades the correlation between the reference and measured spectra. A recursive compensation algorithm is proposed to realize a distributed sensor with higher spatial resolution and precise real-time measurement. This method helps maintain recognizable SNR of correlation by aligning the measured spectra with their reference. Moreover, the more obvious characteristic peak can be located with an accurate strain at the fiber end. With compensation, the distributed strain curve along the 30 m long fiber, at 0.5 mm spatial resolution, approaches that at 2 mm. Besides, the proposed technique can be further developed when combined with parallel computing to increase data capacity and reduce inversion distortion and applied to embedded sensors in the composite materials.

## Figures and Tables

**Figure 1 sensors-20-05540-f001:**
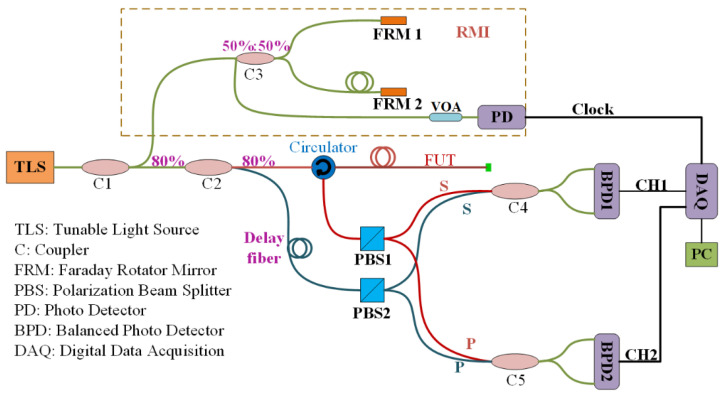
Schematic illustration of the OFDR sensing system.

**Figure 2 sensors-20-05540-f002:**
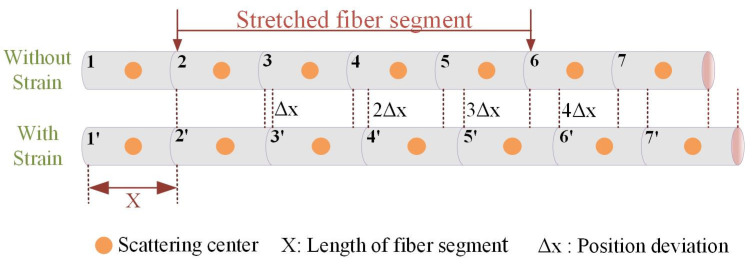
Distribution of scattering center along sensing fiber before and after strain.

**Figure 3 sensors-20-05540-f003:**
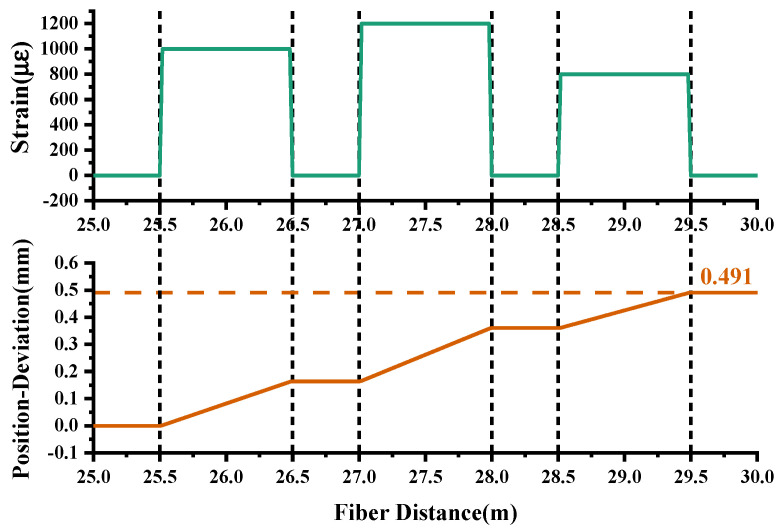
Accumulation of position deviation along sensing fiber after axial strain loaded.

**Figure 4 sensors-20-05540-f004:**
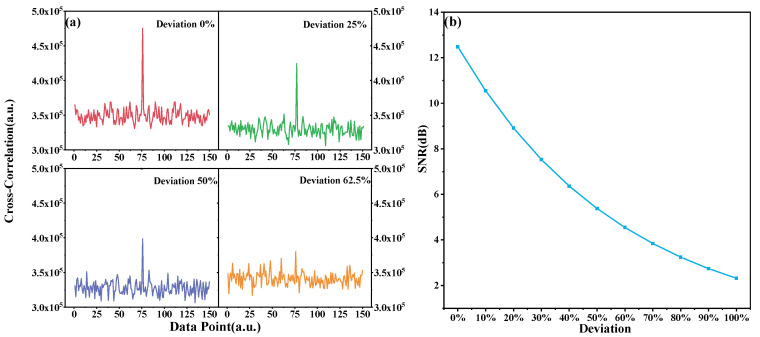
Influence of position deviation on spectral similarity. (**a**) SNR of the characteristic peak to quantify the strength of the cross-correlation. (**b**) SNR decreasing with position deviation.

**Figure 5 sensors-20-05540-f005:**
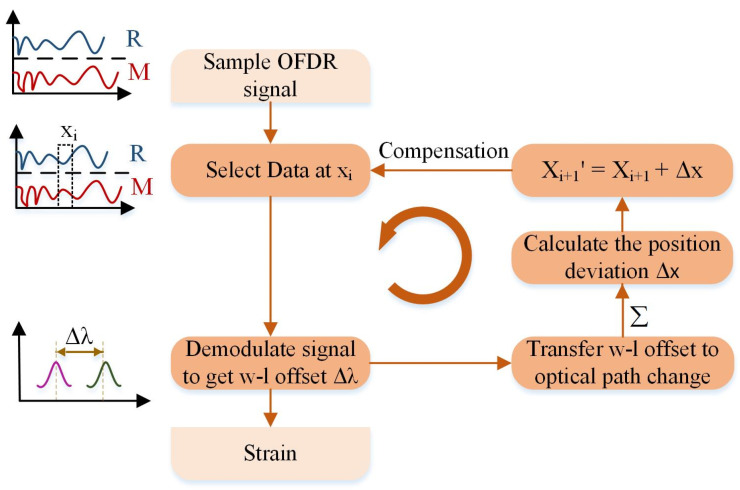
Process of the recursive solution for position deviation to obtain strain distribution.

**Figure 6 sensors-20-05540-f006:**
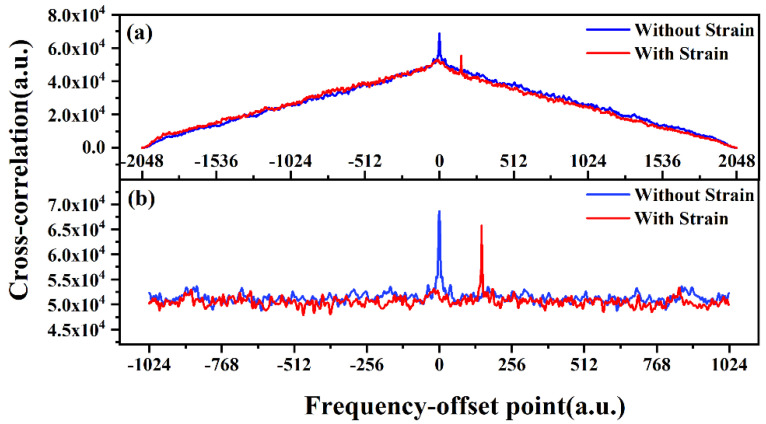
The dependence of the wavelength shift on strain at a single measuring point. (**a**) Cross-correlation in time domain; (**b**) cross-correlation in frequency domain.

**Figure 7 sensors-20-05540-f007:**
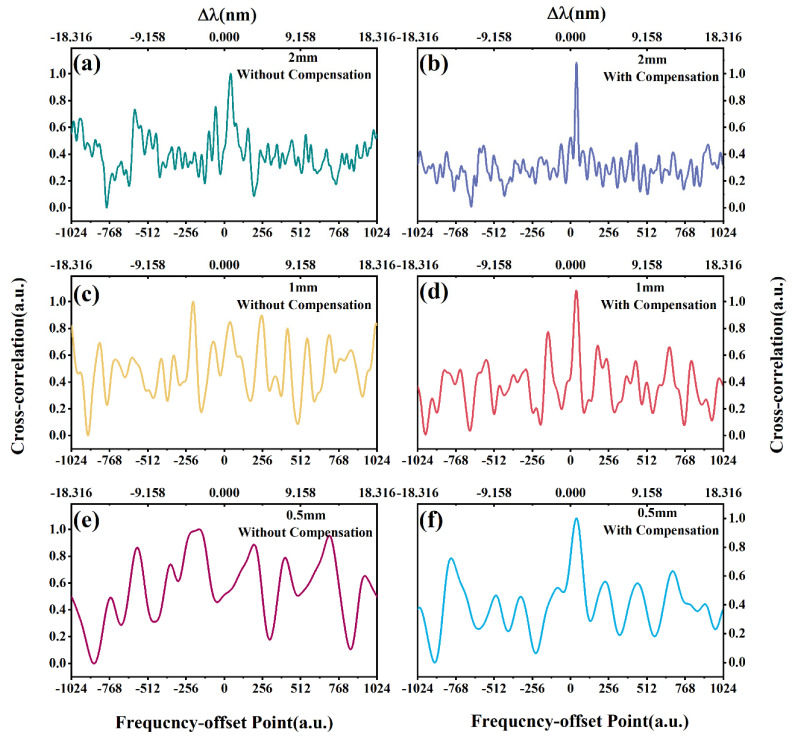
Cross-correlation result of RS spectra before and after position deviation correction: (**a**,**b**) 2 mm spatial resolution; (**c**,**d**) 1 mm spatial resolution; (**e**,**f**) 0.5 mm spatial resolution.

**Figure 8 sensors-20-05540-f008:**
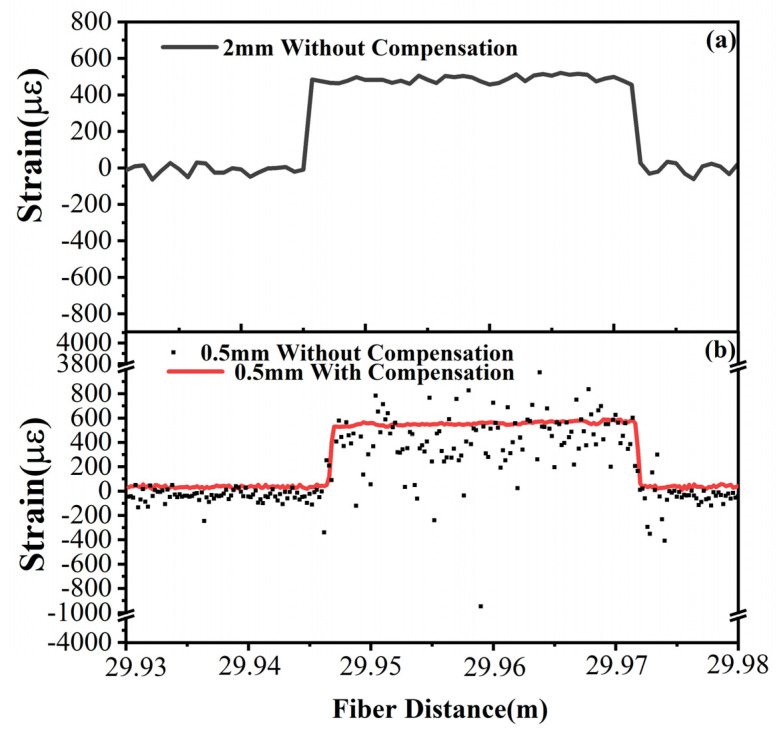
Strain distribution at the fiber end before and after position deviation correction: (**a**) 2 mm spatial resolution; (**b**) 0.5 mm spatial resolution.

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
