# Peer review of "Numerical Analysis and Recursive Compensation of Position Deviation for a Sub-Millimeter Resolution OFDR"

_sensors, 2020, doi:10.3390/s20195540_

Round 1

Reviewer 1 Report

This letter is concerned with the correction of the vibration in front in the sensing of distortion and the like using the optical spectrum correlation in OFDR.

Overall, there not seems to be sufficient explanation and technical originality.

First of all, since there are few precedents of technical examination, we think that it is necessary to explain precisely how much influence of the distortion and in which case correction is necessary. (By the way, there is already a precedent to consider with the OFDR for optical spectrum measurement to dynamic strain measurement, DAS, which is in more complicated situations.)

Although it is explained in FIG. 7, since it is only described at the end of the experimental results, it should be explained in detail at the beginning including the relationship with the resolution.

FIG. 2 illustrates an image of such an example, but the explanation of FIG. 2 is insufficient. Are each of the four points distorted by x? Or is the point in front distorted and quadrupled in the distance? This situation corresponds to all of the background and challenges of this letter, so the value of this letter cannot be understood without a clear explanation of what is happening in this diagram. I think we should detail what's going on.

Figure 4 says that position deviation is 25%, 50%, and 62.5%, but this position deviation should also be well defined. I'm not sure what it means for 0.491 mm displacement. This parameter also affects Figure 4 and should be explained further.

The proposed method is described only by Equation (3) and FIG. 5, and there is no detailed explanation of compensation. What about the first point? Two times measurements? The explanation of the method necessary for using the method is insufficient. Also, the technical novelty of the proposal in the letter is not found much because Equation (3) is so well-known. It is strongly requested that the necessity, accuracy, prospect of improvement in the future, and technical novelty of this method be more strongly added.

On the whole, there are several editorial errors such as the numbers of figures. Full confirmation is required again.

Based on the above, it is considered that substantial additions and revisions are necessary for the publication.

Reviewer 2 Report

The authors analyzed the time varying and accumulative position deviation at the fiber end and demonstrated are cursive compensation algorithm. This method improved the SNR of correlation result and achieved 0.5 mm spatial resolution over 30 m. The working principle was explained, and the sensing performance was well discussed. The manuscript is suitable for publication if the following issues are addressed.

1.The last few sentences in the abstract, introduction and conclusion are very similar and need to be modified.

2.In theory, spatial resolution depends on wavelength sweep range of TLS, but there is not cover the wavelength range of the laser used in your manuscript.

3.As described in your manuscript, Fig. 6a and 6b are the results of correlation in time and frequency domain. What's the difference between these two methods? Can you further explain the counting process?

4. Fig.7 is the cross-correlation results before and after compensation at different spatial resolution. At 1mm and 0.5mm spatial resolutions, after the position deviation corrected, the peak contrast or SNR has no significant improvement. Can you explain this phenomenon?

5.Line 77 in your manuscript, the word “spectrum shift” should be changed into “spectra shift”. The noun is modified by an adjective.

6.The legend in Fig.8 is the same as Fig.7.It needs to be revised.

Author Response

Plesase see the attachment.
